# The Influence of Emotional Environmental Pictures on Behavior Intentions: The Evidence of Neuroscience Technology

**DOI:** 10.3390/ijerph16245142

**Published:** 2019-12-16

**Authors:** Wei-Yin Chang, Ming-Te Lo, Chin-Fei Huang

**Affiliations:** 1College of Forestry, Fujian Agriculture and Forestry University, Fuzhou 350002, China; cwy_forest@163.com; 2Graduate Institute of Science Education & Environmental Education, National Kaohsiung Normal University, Kaohsiung 82444, Taiwan; luoroger@ms28.hinet.net

**Keywords:** electroencephalography (EEG), emotions, environmental intention to act, neuroscience

## Abstract

Background: In recent years, researchers have been paying increasing attention to the issues of how emotions affect people’s perceptions of the environment, and how they influence people’s behavior or intentions to act. The purpose of this study is to explore the influences of emotions on environmental intention to act by using the neuroscience technology electroencephalography (EEG). Methods: A total of 70 university students participated in this study. They looked at positive and negative emotional environmental pictures and decided if they wanted to protect the environment after looking at the pictures. The participants wore an EEG cap throughout the process in order to collect their brain wave (EEG) data. Results: The analysis of variance (ANOVA) results showed that the power value of meditation was significantly higher when the participants looked at the positive than at the negative emotional environmental pictures (*p* < 0.001). The power value of pressure was significantly higher when the participants looked at the negative than at the positive emotional environmental pictures (*p* < 0.001). The power value of attention was significantly higher when the participants looked at the negative than at the emotional environmental pictures (*p* < 0.001). Conclusions and recommendations: The findings showed that positive emotional environmental pictures might promote positive emotions, but will decrease the intention to act to protect the environment. In contrast, negative emotional environmental pictures will increase negative emotions, and will also increase attention and intention to act to protect the environment. Implications of the findings are discussed.

## 1. Introduction

The goal of environmental education has been identified as improving the health of the world’s ecosystems [1]. To teach people to achieve that goal, environmental education practitioners have designed a number of curriculum and teaching strategies to improve general environmental knowledge, attitudes and/or behaviors [2]. As with other subjects, one of the most important components of a teaching strategy for environmental education is emotion [3,4].

In recent years, researchers have been paying increasing attention to the issues of how emotions affect people’s perceptions of the environment [5,6], and how emotions influence people’s behavior or intentions to act [7]. One of the most likely reasons why researchers are focused on the effects of emotions is that emotions are a basic component of human psychology [5], and people will react to the environment depending on their emotions [7,8]. For that reason, environmental practitioners often show pictures or videos which involve emotional components to induce an audience’s positive or negative affect, with the expectation of enhancing the audiences’ intention to protect the environment or decreasing their intention to destroy the environment. Neutral emotions are used as a blank task.

Like emotion, a number of cognitive processes are difficult to explain or detect [6,9]. In these few years, a growing number of researchers have suggested that these kinds of studies on cognitive processes or emotion detection must combine neurophysiological methods [7,10]. To better understand the possible answers to the above-mentioned questions, a neuroscience technology, electroencephalography (EEG), was used in this study.

## 2. Method

### 2.1. Sampling Methods and Sample Size

This study was conducted at an urban university in the south of Taiwan. A total of 70 volunteer university students (35 males, 35 females, average age ± S.D. = 20.34 ± 1.43 years old) participated in the study. The volunteers were recruited by advertising on the campus. The participant information is given in Table 1.

All participants were asked to complete the experimental task by wearing an EEG cap to collect the EEG data. All participants were confirmed to be mentally healthy without a history of neurological or psychiatric disorders, and all gave voluntary consent to participate in the EEG experiments. All participants were given 100 NTD (New Taiwan Dollar) after completing this research.

### 2.2. Procedure and Instrument

The participants were asked to look at a white piece of paper for 1 min to collect blank brain wave data with a neutral emotion. The blank brain wave data were used as the participants’ baseline EEG data. To induce the participants’ emotions, this study adopted two kinds of pictures: 30 positive and 30 negative emotional environmental pictures. The positive and negative emotional environmental pictures were chosen by 120 university students in a pilot study. All of these 120 students were shown 100 environmental pictures and were asked to choose which pictures could induce their positive emotions, negative emotions or no emotions. A picture was defined as positive or negative when more than 80% of the participants agreed, as 80% agreement could ensure the reliability and validity of the instruments. The results of the pilot study showed that the photos did not reach significant differences in gender or age.

According to the results of the pilot study, 30 positive and 30 negative emotional environmental pictures were adopted for use in the formal study (examples are shown in Figure 1) which involved an additional 70 volunteer participants who were not involved in the pilot study. The EEG data were collected from these 70 volunteer participants.

All 70 formal participants in this study needed to look at all 60 pictures individually in a single room with gentle light. The participants were invited to sit on a chair and look at the pictures shown on a computer screen. There was a researcher in the room. If the participants had questions, they could ask the researcher.

After looking at the pictures, participants needed to press a button to decide if they wanted to protect the environment more or not. Then, the participants needed to look at a blank computer screen for 20 s to collect their baseline brain wave (EEG) data. All 60 pictures were randomly shown on the computer screen. Each picture was shown for 20 s. After 20 s, a white page with the words “make a decision” was shown on the computer screen for 10 s. The participants had 10 s to make a decision to press one of the “5”, “4”, “3”, “2”, or “1” buttons. The participants were asked to push the “5” button on the computer keyboard if they felt a strong intention to protect the environment after looking at the picture, while they were asked to push the “1” button of the computer keyboard if they felt the lowest intention. The participants’ responses were collected automatically by the computer when they pushed the button. The procedure is shown in Figure 2.

In this study, the EEG technology (see Figure 3) was used to detect and confirm the participants’ emotional reactions.

This study adopted the EEG signal combinations reflection system which was developed by Sheng Hong Precision Technology Co. Ltd. In this system, the raw EEG signals were collected at 512 data points per second. The raw data can be translated into three kinds of cognitive reactions, namely pressure, attention and meditation, and can also be translated into 8 FFT(Fast Fourier Transform) frequency bands of EEG data which include delta wave (1–4 Hz), theta wave (4–7 Hz), alpha wave (8–14 Hz), beta wave (15–30 Hz), low gamma wave (30–50 Hz) and high gamma wave (>50 Hz).

In past studies, the definition of meditation in EEG detections has been widely accepted as a way to manage stress, where higher meditation data indicate lower stress [11,12]. Therefore, in this study, the definition of “higher pressure” refers to the EEG data of “lower meditation.” Besides, attention means that the participant is focused on one thing and his/her EEG data from the EEG detectors show higher active performance [13]. In this study, the higher power value data of the “attention” item indicated that the participant paid higher attention.

A previous study mentioned that a higher power of high alpha frequency reflects the participants’ positive emotions, and vice versa [11]. Based on the criteria, the participants’ EEG data of high alpha frequency power value were compared to their baseline EEG data to confirm their emotions.

All of the participants’ EEG signals were collected while they were participating in the experiments (see Figure 4), and the signals can be translated into statistical data (the power value) through the EEG technology system.

### 2.3. Data Analysis Methods

The statistical data which were translated from EEG raw signals were entered into SPSS version 23 for analysis. The analysis of variance (ANOVA) was used to analyze the participants’ cognitive reflections of pressure, meditation and attention from looking at different emotional pictures. All factors with a *p*-value < 0.05 in the bivariate analysis are further discussed in this study.

### 2.4. Ethical Considerations

The authors had been trained in the Code of Ethics of the World Medical Association, and the study was approved by the ethics committee of National Cheng Kung University Human Research Ethics Committee.

## 3. Results

The results of the EEG data which were analyzed by analysis of variance (ANOVA) showed that the power value of meditation was significantly higher when the participants looked at the positive emotional environmental pictures (mean ± S.D. = 92.1 ± 3.7) than at the negative pictures (mean ± S.D. = 2.6 ± 2.0) (Table 2).

Besides, the power value of meditation was significantly higher when the participants looked at the positive emotional environmental pictures (mean ± S.D. = 90.2 ± 9.3) than at the negative ones (mean ± S.D. = 6.4 ± 0.5) (Table 3). This indicates that the participants felt higher pressure (lower meditation) when looking at the negative pictures.

However, the power value of attention was significantly higher when the participants looked at the negative emotional environmental pictures (mean ± S.D. = 87.4 ± 8.3) than at the positive ones (mean ± S.D. = 73.6 ± 2.4) (Table 4). The results imply that people will pay more attention when looking at negative than at positive emotional environmental pictures.

Besides, the participants’ behavior data were also collected. In this study, the higher value of participants’ behavior data indicated that the participants had a stronger intention to protect the environment. After looking at a picture, the participants needed to press different buttons to indicate whether they wanted to protect the environment or not. The results from Pearson’s correlation analysis (Table 5) showed that the participants’ behavior responses were significantly and positively more highly related to pressure and attention and significantly and negatively related to meditation. In other words, the participants had a higher intention to protect the environment after feeling higher pressure and attention. A higher pressure and attention could be induced by looking at negative pictures.

## 4. Discussion

This study adopted emotional environmental pictures to induce participants’ emotions and investigated their environmental intention to act after looking at these pictures. EEG technology was used to confirm the participants’ emotional reactions.

The analysis of variance (ANOVA) results of the EEG data indicated that the power values of meditation and pressure showed significant differences between the participants when they looked at the positive and negative emotional environmental pictures. It is reasonable that people will have positive emotions when looking at a positive emotional environmental picture and will feel higher pressure when looking at negative emotional environmental pictures. This finding is similar to that of a previous study [7].

Besides this, Table 5 shows that the participants’ behavior responses were significantly more highly related to emotions. There is a significant negative correlation between meditation emotion and behavior response, which implies that relaxed feelings such as meditation could not induce the participants’ environmental intention to act. In contrast, there is a significant positive correlation between pressure and attention, and behavior response. This result implies that higher pressure and higher attention could induce the participants’ environmental intention to act. Although a previous study investigated the issues of environmental affect and action [2], the current study provides new evidence from neuroscience to illustrate the relationship between environmental affect and environmental intention to act.

Integrating the findings from Table 1, Table 2, Table 3 and Table 4, the results indicate that positive emotional environmental pictures could induce feelings of relaxation, such as meditation. However, positive emotional feedback could not induce strong attention, neither could it induce the intention to protect the environment. Adversely, people might experience pressure by looking at negative emotional environmental pictures, and that pressure could induce their attention and intention to protect the environment. In this study, negative emotional environmental pictures were shown to possibly excite the participants’ reflections and make them decide to do something to help improve the world. Adversely, positive emotional environmental pictures might transmit a message that “it’s all good”, which might decrease the intention to protect the environment.

## 5. Conclusions

In this study, we adopted emotional environmental pictures and EEG technology to explore the participants’ environmental intention to act. The findings indicate that positive emotional environmental pictures might promote positive emotions, but that this will decrease the intention to act to protect the environment. In contrast, negative emotional environmental pictures will increase negative emotions, which will also increase attention and the intention to act to protect the environment.

It is therefore suggested that environmental practitioners carefully choose emotion-inducing materials and strategies in practice. If the aim of the environmental education is to help the audience feel good about the environment, then positive emotional materials and teaching strategies might be suitable. On the other hand, if the aim of the environmental education is to promote the audience’s intention to act to protect the environment, negative emotional materials and teaching strategies might be a better choice.

## Figures and Tables

**Figure 1 ijerph-16-05142-f001:**
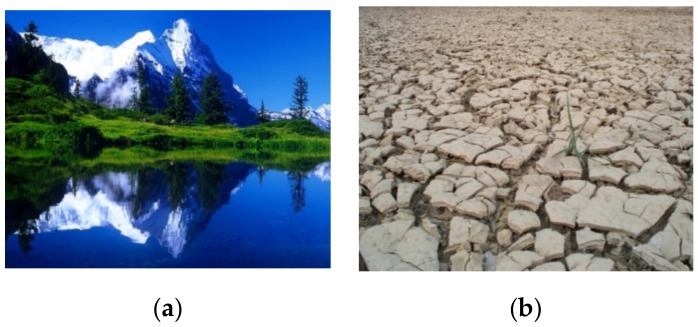
Examples of (**a**) a positive emotional environmental picture, (**b**) a negative emotional environmental picture.

**Figure 2 ijerph-16-05142-f002:**
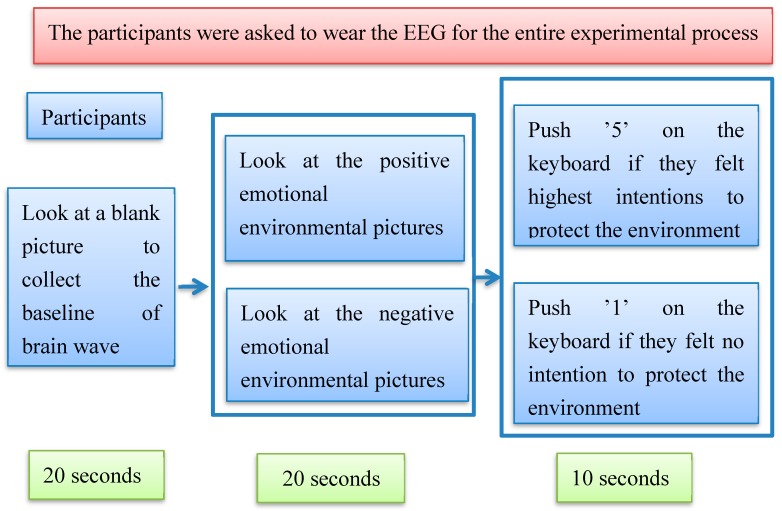
Collection procedures with neuroscience technology.

**Figure 3 ijerph-16-05142-f003:**
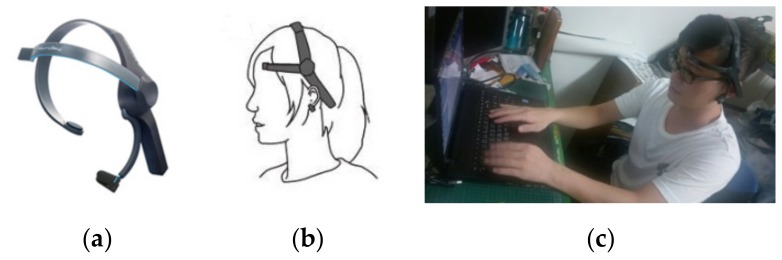
Pictures (**a**) and (**b**) were provided by Sheng Hong Precision Technology Co. (**c**) an example of brain wave collection.

**Figure 4 ijerph-16-05142-f004:**
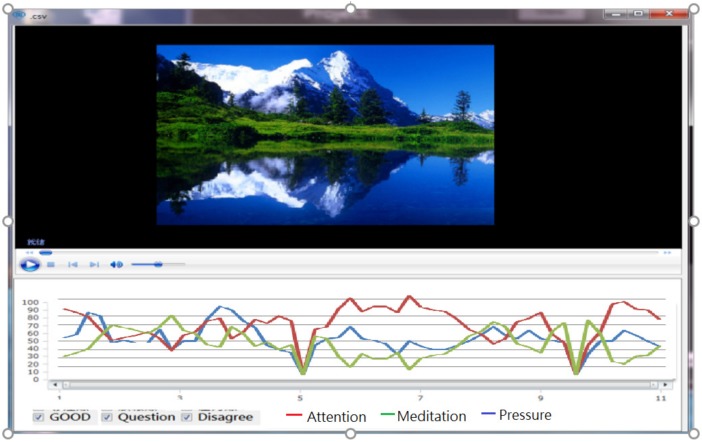
The participants’ neuroscience signals were collected while they were participating in the experiments (see Figure 4), and the signals can be translated into statistical data (the power value) through the neuroscience technology system.

**Table 1 ijerph-16-05142-t001:** The participants’ information.

Information Item	Details
Gender	Males = 35; Females = 35
Age	Average age ± S.D. = 20.34 ± 1.43 years old
Major	Department of Mathematics = 6 Department of Chemistry = 4 Department of Biotechnology = 13 Graduate Institute of Science Education and Environmental Education = 27 Graduate Institute of Adult Education = 1 Department of Education = 3 Department of Chinese = 4 Department of English = 3 Department of Music = 4 Department of Visual Design = 2 Graduate Institute of Interdisciplinary Art = 3

**Table 2 ijerph-16-05142-t002:** ANOVA analysis of the power value of meditation.

Source	df	MS	SS	F
Corrected Model	2	157,98.03	315,96.06	1921.53 ***
Intercept	1	185.32	185.32	22.54 ***
Blank	1	27.72	27.72	3.37
Meditation	1	303,75.10	303,75.10	3694.55 ***
Total	69		765,41.00	

**Note:** SS = sum of squares; MS = mean square; *** *p* < 0.001.

**Table 3 ijerph-16-05142-t003:** ANOVA analysis of the power value of pressure.

Source	df	MS	SS	F
Corrected Model	2	140,94.42	281,88.85	386.43 ***
Intercept	1	1038.02	1038.02	28.46 ***
Blank	1	132.60	132.60	3.64
Meditation	1	280,56.25	280,56.25	769.23 ***
Total	69		659,12.00	

**Note:** SS = sum of squares; MS = mean square; *** *p* < 0.001.

**Table 4 ijerph-16-05142-t004:** The ANOVA analysis of the power value of attention.

Source	df	MS	SS	F
Corrected Model	2	414.69	829.38	12.07 ***
Intercept	1	112,96.57	112,96.57	328.82 ***
Blank	1	73.13	73.13	2.13
Meditation	1	756.25	756.25	22.01 ***
Total	69		1049,60.00	

**Note:** SS = sum of squares; MS = mean square; *** *p* < 0.001.

**Table 5 ijerph-16-05142-t005:** Pearson’s correlation analysis of behavior responses, meditation, pressure and attention.

Source	Behavior Response	Meditation	Pressure	Attention
Behavior Response	1	−725***	699 ***	416 ***
Meditation	−725 ***	1	−951 ***	−345
Pressure	699 ***	−951 ***	1	386 ***
Attention	416 ***	−345	386 ***	1

Note: *** *p* < 0.001.

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
