# Peer review of "The Influence of Emotional Environmental Pictures on Behavior Intentions: The Evidence of Neuroscience Technology"

_ijerph, 2019, doi:10.3390/ijerph16245142_

Round 1
Reviewer 1 Report
raw 38 says “which will be designed ina teaching strategy” must to say “which will be designed in a teaching strategy” raw 64 to 77: The authors are reporting two different studies. The results of the first one is published? How were volunteers recruited? How was selected who participated in the categorization of the photos? Did the 70 volunteers from the neuro perception study also participate in the categorization of the photos? Are there differences in age and sex between both studies? Were they offered money to participate? raw 105 says “If they did not feel lowest intentions to protect the environment” must to say “If they did feel lowest intentions to protect the environment”, raw 122 says “Hz).Prevoius studymentioned that the higherpower of high alpha frequency reflectsthe participants’”, must to say Hz). Previous study mentioned that the higher power of high alpha frequency reflects the participants’” In methodology item author must to define conceptually and operationally way: meditation-pressure- attention and behavior response raw 155 says “Besides, the power value of pressure is significantly higher when the participants look at the negative emotional environmental pictures (Mean ± S.D. =90.2 ± 9.3) than the negative (positive ??) ones (Mean ± S.D. =6.4 ± 0.5) (Table 3)”. Table 3 shows meditation, not pressure. raw 161says “However, the power value of attention is significantly higher when the participants look at the negative emotional environmental pictures (Mean ± S.D. =87.4 ± 8.3) than the positive ones (Mean ± S.D. =73.6 ± 2.4) (Table 4). Table 4 shows always meditation, not attention. Table 5 must be explained in a better way. If meditation pearson´s factor is negative the correlation between meditation and behavior response is and inverse correlation, meanwhile pressure and attention whit positive factors are direct correlation. raws 170 and 172 say “The results from Pearson’s correlation analysis (Table 5) showed that the participants’ behavior responses are significantly higher related to emotions. What kind of emotions? meditation, pressure and attention are emotions? I suggest to analyze the influence of selection bias in the results and to analyze if there are differences or not between males and females.Author Response
Thank you for the encouragement, and thank you for all of your comments and suggestions. They are very helpful for us to enhance this manuscript.

Reviewer 2 Report
This study examined the influences of the emotions on environmental pictures by using neuroscience technology. It's a very interesting study, but one should consider the following:
First, there are various theoretical discussions on the topic of emotion. This study has a limitation that few existing researches on emotion have been reviewed. Considering the type of emotion, not simple emotion, this study adopted very simple approach that gives focus on only one dimension of emotion. Authors should review existing experimental studies on emotion and summarize the contributions of this study.
Second, the experiment uses only one stimulus. There is a need for more sophisticated and elaborated design of experimental designs.
Third, a more detailed explanation of the participants is needed. The environmental awareness of the participants may affect the results of the study, so it is necessary to explain how this part was controlled.
Author Response
Thank you for the encouragement, and thank you for all of your comments and suggestions. They are very helpful for us to enhance this manuscript.

Reviewer 3 Report
In general, the paper shows interesting way to undestand people's emotion from eeg signals. However, there are many aspects which need to be corrected or clarify. Here are the general comments:
Some term is not consistent such as neuroscience cap and brainwave cap.
The author needs to use more standard term such as Electroencephalography (EEG) rather than using the non-standard term like neuroscience cap, brainwave cap, neuroscience technology, neuroscience raw signals.
The procedure is not clearly described. Which measurement point used for EEG measurement? Why the point is used? what is the instrument used in this experiment? what is the specification of the instrument? How is the overall system of recording EEG signal and signal from push button?
The signal processing of the EEG signal Is not clear. Which parameters used from the EEG signals in the analysis? How the author can get the parameters?
There are several typing mistakes in the manuscript.
The statistical reporting is not standard. Consider to use APA style for statistical reporting.
The author needs to do post-hoc analysis after ANOVA.
Suddenly there are analysis about meditation, pressure and attention without previous proper explanation.
Some specific comments:
Abstract
Term neuroscience cap is not standard term. Consider to use more common term such as EEG headset or brainwave sensor
Line 38
Will be design ina teaching..
Line 60
The term brainwave cap is not consistent to the term used in the in abstract (neuroscience cap)
Line 84
What is figure 1?
Line 96
What is neuroscience technology? Why don’t the author used the name/brand of the EEG sensor?
Line 106
Neuroscience technology? What is the meaning of this term?
Line 122
Previous studymentioned
Reflectsthe
Line 135
What is neuroscience raw signals?
Line 136
Which data analysed from the EEG signals? it not clear what parameter used for statistical analysis.
Line 136
ANOVA were used to analyse the participant’s emotion from looking at different emotional pictures. It is not clear what kind of emotional variation used in ANOVA.
Line 172
What is the meaning of this table? How the author gets the behaviour response? What parameter used to represent behaviour response?
Should be Behaviour Response (using capital letter)
Author Response

(The authors gave the same response as above.)

Round 2
Reviewer 2 Report
I did not suggest the publication for this paper.
Reviewer 3 Report
the author has revised the manuscript diligently and finally the experiment has explained in detail.